# Validation of prognostic indices for short term mortality in an incident dialysis population of older adults >75

**Bjorg Thorsteinsdottir**[1,2,3,4]*, **LaTonya J. Hickson**[4,5], **Rachel Giblon**[3,6], **Atieh Pajouhi**[1],
**Natalie Connell**[2], **Megan Branda**[3,7], **Amrit K. Vasdev**[1], **Rozalina G. McCoy**[1,4], **Ladan Zand**[5],
**Navdeep Tangri**[8,9], **Nilay D. Shah**[3,6]

**1** Division of Community Internal Medicine, Department of Medicine, Mayo Clinic, Rochester, Minnesota,
United States of America, **2** Biomedical Ethics Program, Mayo Clinic, Rochester, Minnesota, United States of
America, **3** Knowledge Evaluation Research Unit, Mayo Clinic, Rochester, Minnesota, United States of
America, **4** Kern Center for the Science of Health Care Delivery, Mayo Clinic, Rochester, Minnesota, United
States of America, **5** Division of Nephrology and Hypertension, Department of Medicine, Mayo Clinic,
Rochester, Minnesota, United States of America, **6** Division of Health Care Sciences Research, Mayo Clinic,
Rochester, Minnesota, United States of America, **7** Department of Biostatistics and Informatics, Colorado
School of Public Health, University of Colorado-Denver Anschutz Medical Campus, Aurora, CO, United
States of America, **8** Department of Medicine, Seven Oaks General Hospital, University of Manitoba,
Winnipeg, Canada, **9** Department of Community Health Sciences, Seven Oaks General Hospital, University
of Manitoba, Winnipeg, Canada

* Thorsteinsdottir.bjorg@mayo.edu

doi.org/10.1371/journal.pone.0244081

Magna Graecia di Catanzaro, ITALY

**Data Availability Statement:** The deidentified data
underlying this study is available at Harvard
Dataverse: https://doi.org/10.7910/DVN/KZ8YEL.v

## Abstract

### Rational and objective

Prognosis provides critical knowledge for shared decision making between patients and cli-
nicians. While several prognostic indices for mortality in dialysis patients have been devel-
oped, their performance among elderly patients initiating dialysis is unknown, despite great
need for reliable prognostication in that context. To assess the performance of 6 previously
validated prognostic indices to predict 3 and/or 6 months mortality in a cohort of elderly inci-
dent dialysis patients.

### Study design

Validation study of prognostic indices using retrospective cohort data. Indices were com-
pared using the concordance ("c")-statistic, i.e. area under the receiver operating character-
istic curve (ROC). Calibration, sensitivity, specificity, positive and negative predictive values
were also calculated.

### Setting & participants

Incident elderly (age ≥75 years; n = 349) dialysis patients at a tertiary referral center.

### Established predictors

Variables for six validated prognostic indices for short term (3 and 6 month) mortality predic-
tion (Foley, NCI, REIN, updated REIN, Thamer, and Wick) were extracted from the

**Funding:** This publication was supported by the Mayo Clinic, Robert D and Patricia E. Center for the Science of Health Care Delivery (B.T., R.H., R.G.M, L.J.H), by the Extramural Grant Program (EGP) by Satellite Healthcare, a not-for-profit renal care provider (L.J.H., B.T.), and by the National Institute of Health (NIH) National Institute of Diabetes and Digestive and Kidney Diseases (NIDDK) grant K23 DK109134 (L.J.H.) K23 DK114497 (RGM) and the National Institute on Aging grant K23 AG051679 (B.T.). Additional support was provided by the National Center for Advancing Translational Sciences (NCATS) grant UL1 TR000135. Dr. Shah has received research support through Mayo Clinic from the Food and Drug Administration to establish Yale-Mayo Clinic Center for Excellence in Regulatory Science and Innovation (CERSI) program (U01FD005938); from the Centers of Medicare and Medicaid Innovation under the Transforming Clinical Practice Initiative (TCPI); from the Agency for Healthcare Research and Quality (R01HS025164, R01HS025402, R03HS025517); from the National Heart, Lung and Blood Institute of the National Institutes of Health (NIH; R56HL130496, R01HL131535), National Science Foundation; and from the Patient Centered Outcomes Research Institute (PCORI) to develop a Clinical Data Research Network (LHSNet). Dr. Tangri reports grants and personal fees from AstraZeneca Inc., personal fees from Otsuka Inc., personal fees from Janssen, personal fees from Boehringer Ingelheim and Eli Lilly, grants, and personal fees, and other from Tricida Inc., outside the submitted work. Study contents are the sole responsibility of the authors and do not necessarily represent the official views of NIH. The funders had no role in study design, data collection and analysis, decision to publish, or preparation of the manuscript.

**Competing interests:** Dr.Tangri reports grants and personal fees from AstraZeneca Inc., personal fees from Otsuka Inc., personal fees from Janssen, personal fees from Boehringer Ingelheim and Eli Lilly, grants, and personal fees, and other from Tricida Inc., outside the submitted work. Study contents are the sole responsibility of the authors and do not necessarily represent the official views of NIH or the US government. This does not alter our adherence to PLOS ONE policies on sharing data and materials.

electronic medical record. The indices were individually applied as per each index specifications to predict 3- and/or 6-month mortality.

## Results

In our cohort of 349 patients, mean age was 81.5±4.4 years, 66% were male, and median survival was 351 days. The c-statistic for the risk prediction indices ranged from 0.57 to 0.73. Wick ROC 0.73 (0.68, 0.78) and Foley 0.67 (0.61, 0.73) indices performed best. The Foley index was weakly calibrated with poor overall model fit (p <0.01) and overestimated mortality risk, while the Wick index was relatively well-calibrated but underestimated mortality risk.

## Limitations

Small sample size, use of secondary data, need for imputation, homogeneous population.

## Conclusion

Most predictive indices for mortality performed moderately in our incident dialysis population. The Wick and Foley indices were the best performing, but had issues with under and over calibration. More accurate indices for predicting survival in older patients with kidney failure are needed.

## Introduction

Optimal shared decision making is predicated on informed and evidence-based conversations between the patient, caregiver, and clinician. For people with end stage renal disease (ESRD) the decision about pursuing renal replacement therapy (RRT) requires a clear understanding of the differences in prognosis with initiation of dialysis, pursuit of kidney transplantation, or maintenance of conservative therapy [1, 2]. This conversation is particularly important for patients for whom dialysis is a destination therapy, and whose prognosis while receiving dialysis may be poor [3–5]. Many nephrologists and primary care clinicians, hesitate to share prognostic information with patients [6] and feel unprepared for discussions about prognosis and goals of care [6–9]. This hesitancy stems, in part, from lack of a commonly accepted and widely used standard for predicting and communicating prognostic information to patients and caregivers. Absence of real-time prognostic guidance may contribute to the current default to pursue more aggressive treatment options and deprive patients of the opportunity to make informed choices about their health and healthcare [10–12].

The rate of incident ESRD is highest among older adults [13], with high treatment and symptom burden [14] resulting on average in 44.2% of older patients dying within first six months of dialysis initiation [5]. Once on dialysis upward of 50% of elderly patients choose to withdrawal treatment before death [15]. Several prognostic indices have been developed to predict mortality in dialysis patients [16–26]. However, there has been limited uptake of these tools into routine clinical practice and limited research of their utility and impact for shared decision making especially in the oldest patients. The available indices have variable performance with most have moderate to good accuracy in development cohorts that do not always hold in external validation.

Better understanding of the generalizability, performance, and advantages/disadvantages of the available prognostic mortality indicators is needed to assess their utility in real-world populations. The primary aim of the study was therefore to examine the performances of the available prognostic indices in a cohort of elderly (aged 75 years and older) patients newly initiated on RRT.

## Methods

This was a prognostic index validation study, following the TRIPOD checklist for prediction model validation [27].

### Study design and population

The cohort included all adults aged 75 years and older who initiated any type of RRT from January 1, 2007, through December 31, 2011 in the Mayo Clinic Dialysis Services (MCDS) which provides all RRT services in our health system and serves a general population of 385,000 patients in Southeast Minnesota, Northern Iowa, and Southwest Wisconsin, through 8 community based HD facilities as well as inpatient HD. Patients were excluded if they did not provide the institutions generic research authorization, in accordance with Minnesota state law, or if they initiated RRT at another institution or if they had previously received a kidney transplant. Mayo Clinic Institutional Review Board reviewed and approved this study. The de-identified study dataset can be made available upon request from the corresponding author.

### Prognostic indices

We identified 11 indices validated for use at RRT initiation, predicting short term survival (3–6 months), through a systematic review of mortality prediction indices [16]. We had the necessary data to calculate 6 of the indices, three (Foley, REIN and NCI) [18, 22, 28] had been previously validated externally, whereas for the other three (Updated REIN, Wick and Thamer) [19, 24, 29], this paper serves as the first external validation. The indices were developed and tested in cohorts of different, size and composition general vs. geriatric and varied in their inclusion or exclusion of patients with acute kidney injury AKI (Table 1). Most had a c-statistic around 0.7–0.8 in development and internal validation but varied in their performance in previous external validation [16].

### Primary outcome

Primary outcomes were index discrimination as measured by the c-statistic or the area under the receiver operating characteristics curve (ROC); calibration as measured by the Hosmer-Lemeshow; goodness of fit statistic and calibration curves; and positive and negative predictive value to predict 3- and 6-month all-cause mortality.

### Independent variables

Data on patient demographics (sex, marital status, and living arrangement), comorbidities, context, and survival was extracted from the EHR by a college student supervised by an internist and nephrologist (BT, LJH). Living arrangement was classified as independent and assisted living and nursing home (NH). Comorbidities extracted manually from past medical history were supplemented with a validated electronic search from the EHR that was then used to calculate the Charlson Comorbidity Index (CCI) [30]. Functional status for hospitalized patients was based on the Barthel's index was calculated by a validated electronic search pulling information from nursing assessment for hospitalized patients [31]. For patients without a

**Table 1. Demographics for current population and development populations for the different indices.**

| | MCDS cohort | Foley | NCI | REIN | Updated REIN | Thamer | Wick |
|---|---|---|---|---|---|---|---|
| | N = 349 | N = 325 | N = 21043 | N = 2500 | N = 12500 | N = 52796 | N = 2199 |
| **Age, years (mean, SD)** | 81.5 (4.4) | — | — | 80.9 (4.1) | — | 76.9 (6.5) | 75.2 (6.5) |
| **Age, years (N, %)** | 75+ | 18+ | 65+ | 75+ | 75+ | 67+ | 65+ |
| < 70 | — | 255 (78.5) | 7024 (33.4) | — | — | — | 576 (26.2) |
| 70–74 | 1 (0.3) | 70 (21.5) | 6406 (30.4) | — | — | — | 556 (25.3) |
| 75–79 | 154 (44.1) | | 4568 (21.7) | 1192 (47.7) | 5103 (41.0) | — | 529 (24.5) |
| 80–84 | 116 (33.2) | | 2224 (10.6) | 925 (37.0) | 4549 (36.5) | — | 528 (24.0) |
| ≥ 85 | 78 (22.4) | | 821 (3.9) | 383 (15.3) | 2801 (22.5) | — | |
| **Gender (N, %)** | | | | | | | |
| Male | 230 (65.9) | 211 (64.9) | 9526 (45.3) | 1509 (60.4) | 7549 (60.4) | 28422 (53.8) | 1336 (60.8) |
| **Race (N, %)** | | | | | | | |
| White | 330 (94.6) | — | — | — | — | 39794 (75.4) | 2122 (96.5) |
| Black | 2 (0.6) | — | — | — | — | 10545 (20.0) | — |
| Other/missing | 17 (4.9) | — | 21043[5] | — | — | 2504 (4.7) | 77 (3.5) |
| **Functional status (N, %)** | | | | | | | |
| Independent living/walks unassisted | 241 (69.1) | — | — | 1673 (66.9) | 7355 (70.5) | — | — |
| Assisted living/needs assistance for ADL or transfers | 12 (3.4) | — | — | 619 (24.7) | 2316 (22.2) | 11108 (21.0) | — |
| NH/total dependency | 34 (9.7) | — | — | 208 (8.3) | 709 (7.3) | — | — |
| Other/missing | 62 (17.8) | — | — | — | — | — | — |
| **Comorbidities (N, %)** | | | | | | | |
| CHF | 114 (32.7) | 122 (37.5) | 6450 (30.7) | 949 (38.0) | 3960 (34.8) | 27701 (52.5) | 1143 (52.0) |
| Sepsis | 65 (18.6) | 9 (2.8) | — | — | — | — | — |
| CAD/ASHD | 129 (37.0) | 112 (34.5) | 6505 (30.9) | 879 (35.1) | 3835 (32.8) | 27272 (51.7) | — |
| CVA/TIA | 30 (8.6) | — | 3418 (16.2) | 311 (12.4) | 1549 (13.2) | — | 579 (26.3) |
| PVD | 44 (12.6) | 19 (5.9) | 1173 (5.6) | 746 (29.9) | 2663 (23.5) | 14550 (27.6) | 265 (12.1) |
| COPD | 38 (10.9) | — | 3077 (14.6) | 335 (13.4) | 1739 (14.9) | 14806 (28.0) | 883 (40.2) |
| Liver Disease | 1 (0.3) | 3 (0.9) | 1658 (7.9) | 22 (0.9) | 126 (1.1) | — | 38 (1.7) |
| Dysrhythmia | 150 (43.0) | 38 (11.7) | 2184 (10.4) | 799 (32.0) | 3916 (33.3) | — | 541 (24.6) |
| Cancer | 100 (28.7) | 20 (6.2) | 1751 (8.3) | 231 (9.2) | 1487 (12.6) | 7423 (14.1) | 287 (13.1) |
| Diabetes | 98 (28.1) | 64 (19.7) | 10915 (51.9) | 933 (37.3) | 4871 (40.4) | 30843 (58.4) | 1275 (58.0) |
| Hypertension | — | 55 (16.9) | — | — | — | — | 2049 (93.2) |
| Smoker | 12 (3.4) | 55 (16.9) | — | — | — | 1841 (3.5) | — |
| **Weight mean (SD)** | 82.6 (18.2) | — | — | — | — | — | — |
| **BMI, kg/m² mean, (SD)** | 28.8 (5.7) | — | — | — | — | 28.0 (6.9) | — |
| **BMI, kg/m² N, (%)** | | | | | | | |
| <18.5 | — | — | — | 164 (6.6) | 430 (4.6) | — | — |
| 18.5–25 | — | — | — | 1232 (49.3) | 4370 (46.9) | — | — |
| ≥25 | — | — | — | 1103 (44.1) | 4510 (48.5) | — | — |
| **Hemoglobin mean, (SD)** | 10.3 (1.6) | — | — | — | — | 10.0 (1.5) | — |
| **Serum albumin, g/dl (mean, SD)** | 3.3 (0.6) | < 3g/dL 65 (20.0%)[3] | — | < 25 g/L 864 (9.3) | — | 3.2 (0.65) | — |
| **Serum Phosphate** | 5.2 (1.8) | — | — | — | — | — | — |

(*Continued*)

**Table 1.** (Continued)

| | MCDS cohort | Foley | NCI | REIN | Updated REIN | Thamer | Wick |
|---|---|---|---|---|---|---|---|
| | **N = 349** | **N = 325** | **N = 21043** | **N = 2500** | **N = 12500** | **N = 52796** | **N = 2199** |
| **Serum creatinine** | 3.8 (2.0) | — | — | — | — | — | — |
| **eGFR (ml/min/1.73 m$^2$)** | 15.0 (14.0) | — | — | — | — | 12.1 (5.1) | — |
| **Formula used for GFR calculation** | CKD-EPI | — | — | — | — | CKD-Epi | CKD-Epi |
| 0–9.9 | 103 (29.5) | — | 1330 (60.5) | — | — | — | — |
| 10–14.9 | 72 (20.6) | — | 434 (19.7) | — | — | — | — |
| ≥ 15 | 174 (49.9) | — | 435 (19.8) | — | — | — | — |
| **Barthel score (mean, SD)** | 83.3 (23.9) | — | — | — | — | — | — |
| **Mortality (N, %)** | 144 (41.3) | 73 (22.5) | 11272 (53.6) | 470 (18.8) | 2548 (10.5)[2] | 6477 (12.3) | 375 (17.1) |
| **Unplanned dialysis start** | ESKD 89 (25.5) / AKI 75 (21.5) / Acute on Chronic 185 (53.0) | CKD 196 (60%) / Acute on Chronic 108 (33%) / AKI 21 (7%) excluded potentially reversible AKI | — | 859 (34.4%) | 31% | — | Excluded AKI |
| **Hospitalization at dialysis start** | 271 (77.6) | — | — | — | — | No of hospitalizations and total hospital days in the prior six months | Hospitalization in the prior 6 months / No 1,242 (56.5) / Yes 957 (43.5) |
| **RRT modality n (%)** | HD 99% / PD 3 (0.9) | — | HD 20283 (96.4) / PD 759 (3.6) | — | — | 50568 HD 95.8% | 1,881 (85.5) HD; / 318 (14.5) PD |
| **Vascular access** | Catheter 199 (75.1) / Fistula 66 (24.9) / Graft 5 (1.9) / PD cath 3 (0.8) / Other 1 (0.3) | — | — | — | — | Catheter 31970 (60.6%) | — |
| **Country** | USA | Canada | Taiwan | France | France | USA | Canada |
| **3 and 6 months mortality** | 142 (40.2%) | 73 (22.4%) | — | 19% | 10.5% / — | 12/3% / 20.3% | 375 (17.1%) |

Abbreviations: NH, Nursing Home; CHF, Congestive Heart Failure; CAD/ASHD, Coronary Artery Disease/Atherosclerotic Heart Disease; CVA/TIA, Cerebro Vascular Accident/Transient Ischemic Attack; PVD, Peripheral Vascular Disease; COPD, Chronic Obstructive Pulmonary Disease.

1. Patient demographics are based on training data.

2. Overall mortality reported for both training and validation data sets, N = 24348.

3. Frequency of those with albumin < 3 g/dl.

4. Calculated; results were reported separately for survival groups.

5. Presumed mostly Asian.

hospitalization, we used patient provided information (PPI) of functional status obtained from an annual questionnaire completed by patients as part of routine care in the outpatient setting (S1 Table). Baseline data were collected on the closest available data prior to dialysis extending back up to 30 days for laboratory values, 1 year for outpatient functional status and 2 years for comorbidities. Laboratory results for hemoglobin, creatinine, CRP, phosphorous and albumin

were pulled from the EHR. GFR was calculated using the CKD-EPI equation [32]. Mortality and death dates were identified by an EHR review through December 27, 2013 and were supplemented with online queries for publicly available death certificates and obituaries for each individual patient based on name and date of birth.

## Statistical analysis

We compare and contrast descriptive statistics of the study cohort to those used by each of the prognostic indicator development study, with the exception of NCI for which we used data from a validation cohort in a study focused on elderly incident RRT patients [28]. Data for all variables used in the prognostic indices are presented as means and standard deviations for continuous variables, and counts and frequencies for categorical variables.

A score for each patient for each of the six prognostic instruments was calculated based on original model parameters specified in their respective development papers. Categorization into high and low risk groups also followed the classifications the original papers. A separate logistic regression model was run for each of the indices to predict death at 3 and/or 6 months post RRT initiation using the prognostic score as the independent variable. Indices were compared using the concordance ("c")-statistic, corresponding to the ROC; higher c-statistic indicates a better preforming model. Sensitivity, specificity, positive and negative predictive values, and positive and negative likelihood ratios were also calculated. We created calibration plots to evaluate predicted probability of death vs. true observed mortality rate (true probability) in R using the "Presence-Absence" package. A Hosmer-Lemeshow goodness of fit test was performed for each index to assess whether differences between the observed and expected proportions of the outcome were significant, indicating poor model fit. The investigators and analysts were not blinded to the, tools, predictors or outcome.

## Missing data

Complete data was available to implement three of the six indices (Foley, NCI and Wick). For the remaining three indices (REIN, Updated REIN, Thamer) almost 52% of patients were missing at least one variable, ranging from 3% missing the Barthel score to 38% missing the BMI. For missing BMI, albumin, and Barthel score data, we first tested the assumption that variables were missing at random (by testing for collinearity and interaction with other variables) and then imputed by means of multiple imputation using chained equations (10 replications) in STATA. Prognostic scores were generated for each imputed data set separately and averaged over the fitted indices. Analysis was preformed using STATAMP version 15.1 (StataCorp, LP), and R 3.4.2.

## Results

The patient population of 349 older adults initiating RRT was, on average, 81.5±4.4 years old, 66% were male, 94.6% were non-Hispanic white (Fig 1, Table 1). This cohort was smaller and older than most of the studies and most comparable to the REIN cohorts in terms of age and functional status. The overall burden of comorbidity was high, with coronary artery disease (CAD), chronic heart failure (CHF), and diabetes being the most common. Median survival was 351 days with 132 patients dying before 90 days (37.8%) and 142 (40.6%) before 6 months. Sixty patients (17%) recovered renal function and discontinued RRT during the follow up, they were not censored as our interest was in overall survival. Functional status was similar between patients who started inpatient (Barthel score 83.6 +/- 22.3) vs. outpatient (84.5 +/- 22.2).

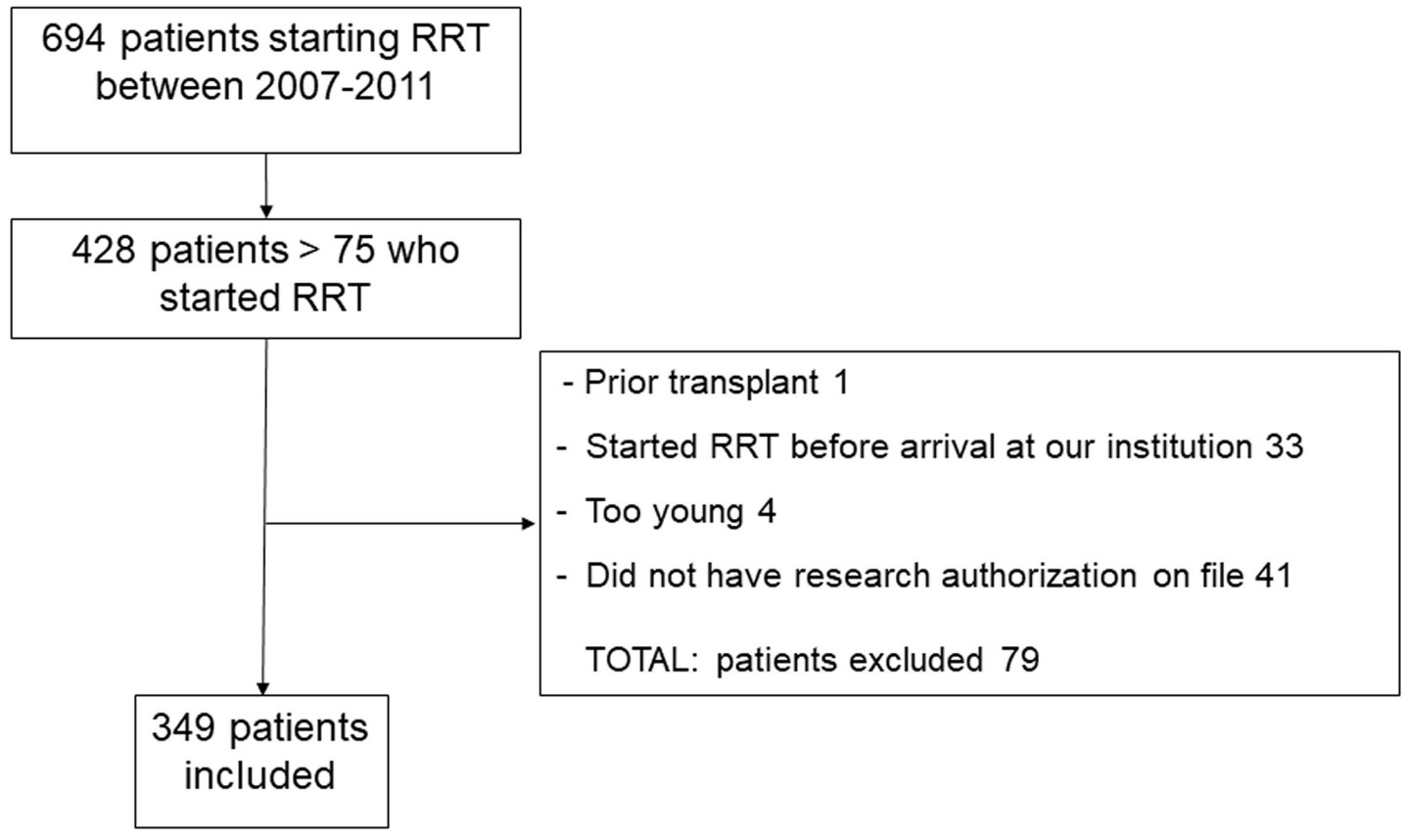

**Fig 1. Flowchart of cohort development.**

With the different indices of interest using different variables and predicting different levels of risk, the resulting risk stratification of our cohort varied depending on which index was used (Fig 2, Table 2). The "high risk" designation was assigned by 22.6% of our cohort when the Foley index was used, compared to 0.9% of the cohort with the Thamer index. This was not necessarily consistent with the predicted mortality threshold corresponding to "high risk" of death, since "high risk" in the Foley index corresponds to 90–100% 6-months mortality, whereas it is >55% 6-months mortality for the Thamer index (Table 2).

None of the indices performed well in our index with only Wick having ROC >0.7, at 0.73 (95% CI: 0.68, 0.78). A comparison of ROCs across all 6 indices indicated that they did not substantially differ in their predictive ability (Table 3, Fig 3). Predicted mortality for four (REIN, NCI, Wick, and Thamer) underestimated mortality for the highest risk group, while Foley markedly overestimated it (Table 2). Table 3 shows positive and negative likelihood ratios with Thamer, Wick, and Foley indices performing best.

Calibration plots for each of the indices are shown in Fig 4. For the two indices predicting 3-month mortality (Updated REIN and Thamer), 3-month predictions were slightly better calibrated than their 6-month counterparts. Of the two indices with highest discrimination, the Foley index was weakly calibrated with poor overall model fit (p <0.01), while the Wick index was relatively well-calibrated.

Using the pre-specified cutoffs for "high risk" defined by each index, the PPV for mortality in the high risk group ranged from 41.9% to 62.4% (Wick performed the best), and NPV ranged from 0–100% (REIN and Thamer performed the best).

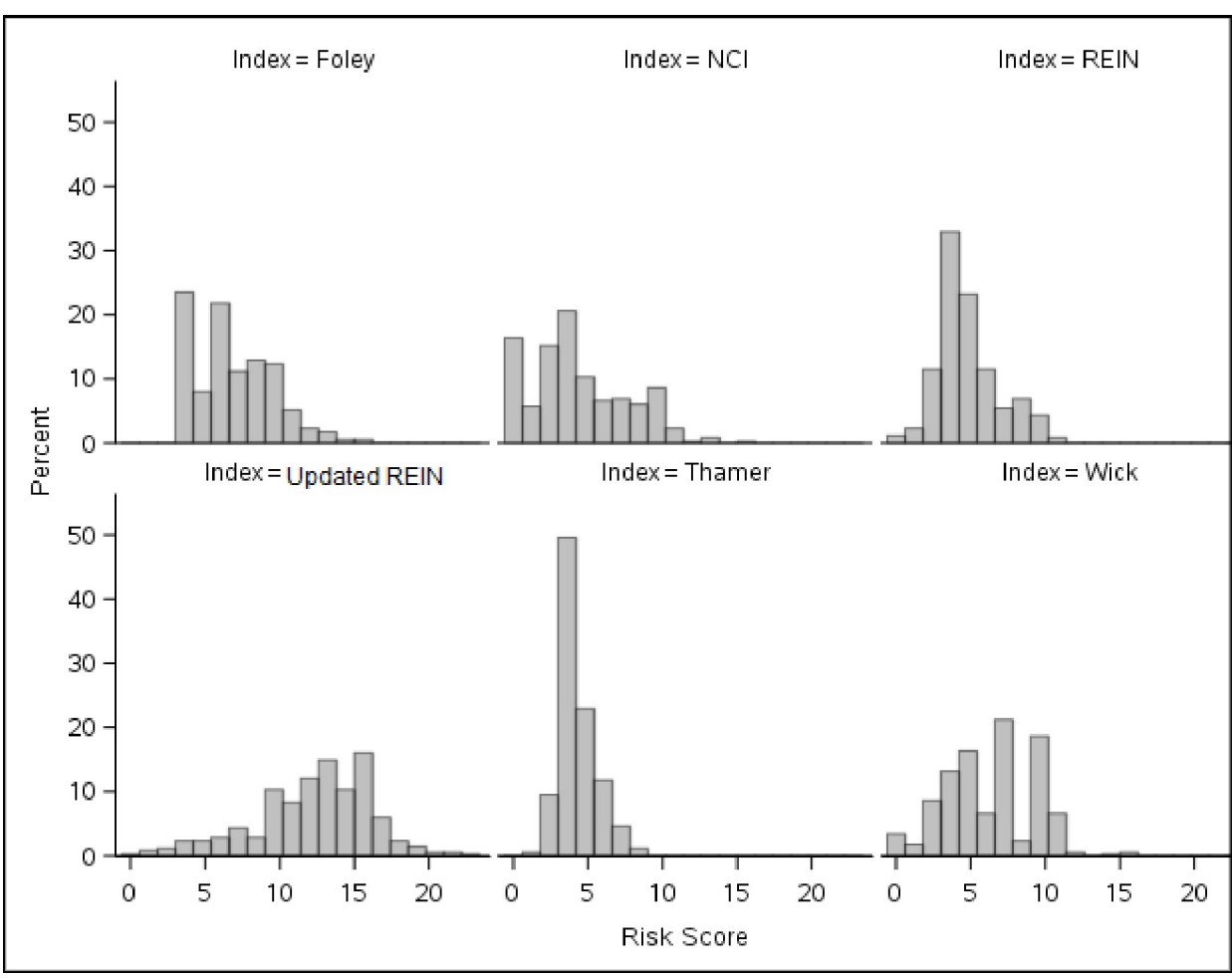

**Fig 2. Percent of patients with each risk score value, by index.**

To improve the performance of these indices, we identified different risk thresholds for each index that would be optimized for our patient population. The following cutoff scores yielded a specificity of >50% and >90%, respectively, in predicting mortality: Foley 7 and 10, NCI 4 and 10, REIN 4.2 and 8.2, updated REIN 12.1 and 16.4, Thamer 4.2 and 6.5, Wick 6 and 10.

## Discussion

Prognostic information is desired by patients and can facilitate and improve shared decision making [9, 33]. We tested six indices predicting short-term (3- or 6-month) mortality at the start of RRT [16]. Their performance in our population-based cohort of elderly incident RRT patients was variable. The discrimination, which reflects the probability that a randomly selected patient who died had a higher risk score than a patient did not die, was poor for all indices except for Wick, which had good discrimination ROC 0.73. Calibration, i.e. the agreement between observed and expected (i.e. predicted) outcomes, was acceptable only for the Wick and Thamer indices. All of the indices fell short in their ability to predict death for the highest risk group. Most concerning was the low positive and high negative predictive values of all the prognostic indicators in the highest risk patient subgroups, as this may lead patients

**Table 2. Breakdown of cohort into predicted risk categories[1] by index, actual and predicted mortality (%) at 3 months and 6 months after dialysis initiation by risk score.**

| Index Breakdown of MCDS cohort N (%) | Score | Points | Predicted 3 mo. mortality | Actual 3 mo. mortality | Predicted 6 mo. mortality | Actual 6 mo. mortality |
|---|---|---|---|---|---|---|
| **Foley** | | | | | | |
| 79 (22.6) | High | ≥9 | — | 55.7 | 90–100% | 59.5 |
| 188 (53.9) | Moderate | 5–8 | — | 39.4 | 33–47% | 43.6 |
| 82 (23.5) | Low | <5 | — | 17.1 | 4% | 18.3 |
| **NCI** | | | | | | |
| 27 (7.7) | High | ≥10 | — | 44.4 | 23.7–38.4% | 48.2 |
| 165 (47.3) | Moderate | 4–9 | — | 47.9 | 11.4–32.3% | 50.3 |
| 157 (45.0) | Low | <4 | — | 26.1 | 6.5–30.6% | 30.6 |
| **REIN** | | | | | | |
| 18 (5.2) | High | ≥9 | — | 44.4 | 25% | 55.6 |
| 161 (46.1) | Moderate | 5–8 | — | 46.0 | 9–15% | 49.1 |
| 170 (48.7) | Low | <5 | — | 29.4 | 3% | 32.4 |
| **Updated REIN[1]** | | | | | | |
| 34 (9.7) | High | ≥17 | > 40% | 41.2 | — | 47.1 |
| 187 (53.6) | Moderate | 12–16 | 20–40% | 44.4 | — | 47.6 |
| 128 (36.7) | Low | <12 | < 20% | 27.3 | — | 30.5 |
| **Thamer** | | | | | | |
| 3 (0.9) | High | ≥8 | 39% | 100.0 | > 55% | 100.0 |
| 116 (33.2) | Moderate | 5–7 | 22–34% | 43.1 | 35–49% | 46.6 |
| 230 (65.9) | Low | <5 | 2–17% | 34.4 | 4–27% | 37.8 |
| **Wick** | | | | | | |
| 31 (8.9) | High | ≥10 | — | 67.6 | > 50% | 72.0 |
| 224 (64.2) | Moderate | 4–9 | — | 46.0 | 25–50% | 48.2 |
| 94 (26.9) | Low | <4 | — | 8.5 | < 25% | 14.9 |

1. As defined by original paper for each index, if there were more than 3 categories defined in the paper we took the lowest and highest categories and collapsed the other categories into moderate.

and clinicians to forego life-sustaining treatment due to underestimation of life expectancy and potential benefit. The indices performed considerably better in predicting survival for the lowest risk patients. Thus, they may be more helpful to promote optimism and treatment options such as dialysis and kidney transplant for patients with reasonably good chances of survival.

The indices that performed best in our elderly cohort included functional status and hospitalizations in the last 6 months, as well as proxy variables suggestive of unplanned dialysis start, all three are important markers of poor health or sentinel events [34–37]. The other indices variably included similar variables but not all three. While disappointing, the c-statistics for the different indices in our validation study are similar to those reported in multiple other validation studies summarized in our recent systematic review and thus can also be seen to show reasonable reproducibility of the initial studies [16]. It is not unusual for prognostic indices to perform worse in a new population than in the development cohorts and our findings again demonstrate how difficult it is to develop completely accurate and reliable models that are generalizable to different settings of a heterogeneous patient population. When the Foley index was initially validated it did poorly, the discrimination of the REIN index ranged from 0.68–0.74 in the initial development and validation study and has varied from 0.66–0.70 in

**Table 3. Discrimination, calibration and predictive values.**

| Index | Discrimination Area Under the Curve | Calibration HL-goodness of fit | PPV for high risk score | | Likelihood ratio + for high risk predicted to die | NVP for low risk score | Likelihood ratio–for low risk predicted to live |
|---|---|---|---|---|---|---|---|
| Foley | 0.67 (0.61, 0.73) | 0.004 | ≥9 | 59.5% | +2.09 | 81.7% | -0.32 |
| NCI | 0.63 (0.58, 0.69) | 0.23 | ≥10 | 51.7% | +1.32 | .% | -0.43 |
| REIN | 0.61 (0.55, 0.67) | 0.45 | >9 | 49.7 | +1.78 | 100% | -0.54 |
| Updated REIN | 0.62 (0.56, 0.68) | 0.03 | ≥17 | 41.9% | +1.27 | 66.7% | -0.62 |
| Thamer | 0.57 (0.51, 0.63) | 0.43 | ≥8(≥7) | 42.8% | +2.44 | 100% | -0.32 |
| Wick | 0.73 (0.68, 0.78) | 0.70 | >12 | 62.4% | +2.14 | 85.1% | -0.25 |

(HL–Hozmer- Lemeshow goodness of fit).

PPV—positive predictive value for highest risk group (expected to die).

NPV–negative predictive value for mortality and lowest risk group (expected to survive).

external validation studies and external validations of the NCI from 0.60–0.91 [16, 18, 19, 22, 24, 28, 29, 38]. Our findings are however lower than those reported by Ramspek et al. [39] in a recent validation study. Their study looked at 1 year prognosis and thus included a different set of prognostic indices with the Foley index being the only one included in both studies. We were unable to include the two best performing indices in the Ramspek study because they included variables not available at dialysis start including dialysis adequacy and treatment modality after 3 months on dialysis [25, 26]. In addition to their strength of size and generalizability of a population based cohort, the difference in discrimination may also tie to the fact that we looked at mortality from the day of RRT initiation whereas they gathered baseline data and started the prediction validation at day 90 of RRT. Thus our mortality rate was significantly higher than the other studies as well as the mortality in our population when limited to patients who survive the initial 30 days of HD [40]. While this has long been customary for studies on ESKD to ensure that patients do in fact have ESKD as opposed to acute kidney injury, we feel this fails to help patients and their clinicians make decisions at the time of dialysis initiation and fails to account for the high early mortality [5].

The lack of generalizability of the examined indices likely stem from the varied populations in which these indices were developed and differences in predictive variables chosen and may reflect overfitting to the development populations. Our population differed by representing a narrower age range with a higher mortality rate than reported in most of the development studies. If age were appropriately factored into the models however, then applying their weights should yield accurate results. Also a well calibrated index should be able to perform in new populations with higher and lower mortality rates than the original development populations. We acknowledge that the small size of our cohort contributes to the poor fit of the indices, but is representative of the difficulties likely faced by other health care organizations with a limited number of patients with incident ESRD. The event rate for our primary outcome of 6 month mortality was approximately 40%; thus, we were sufficiently powered to assess all of the indices.

Moreover, our cohort is limited by its small size, the racial homogeneity of our cohort as mostly white also contrasts with the general US ESKD population but it is unclear how it

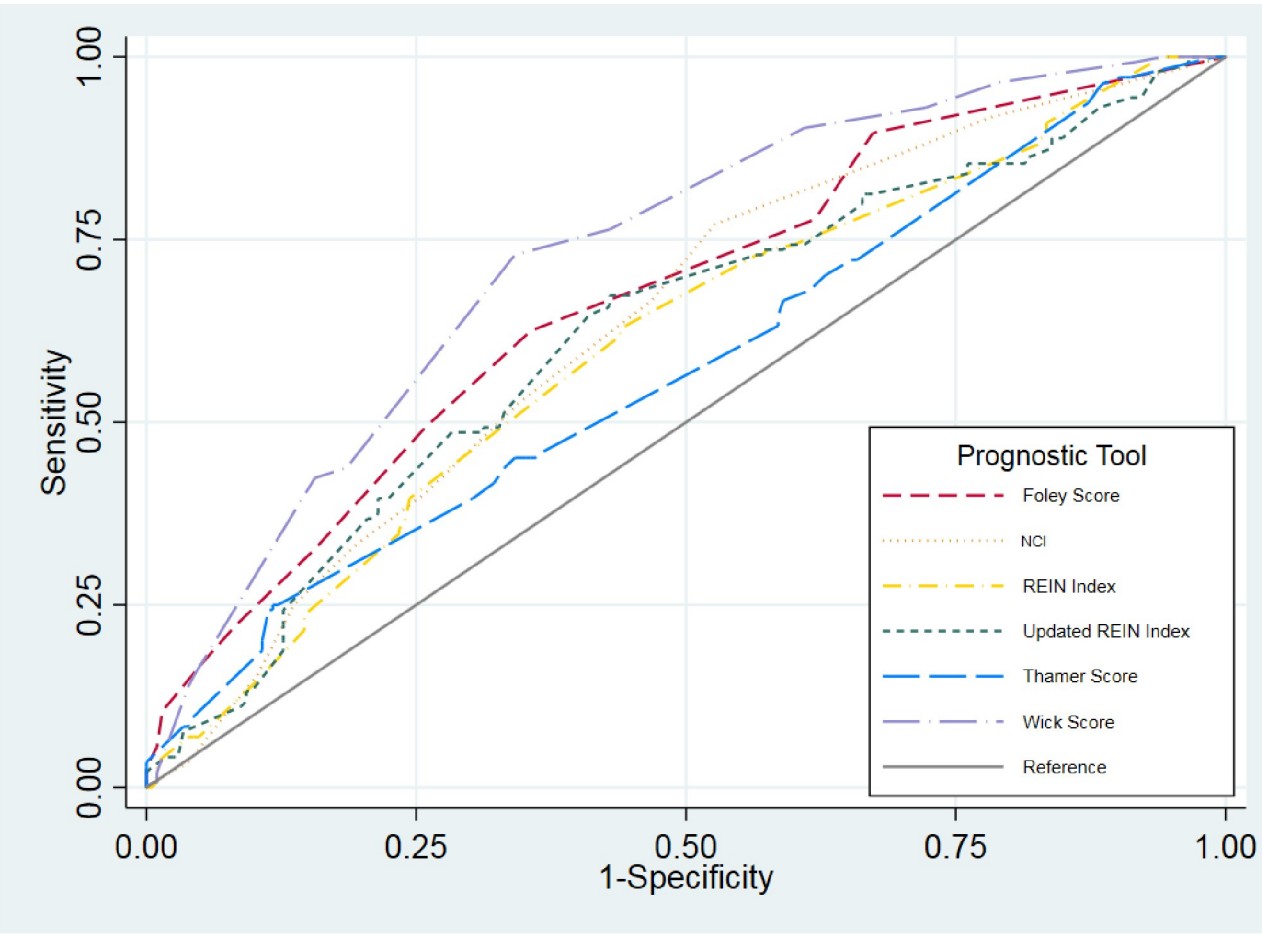

**Fig 3. Receiver operating characteristics (ROC) curves for each index.**

compares to the populations used in most previous development and validation studies that often did not report on race [16]. Another limitation is the use of secondary EHR data which is only as good as the initial documentation allows. We did supplement manual extraction with validated algorithms for data extraction for important variables as well as imputation for key missing variables necessary for the construction of the index scores. When imputing we did make sure that our data suggested that they were missing at random. We did use different methods to assess functional status for inpatients (nursing assessment) vs. outpatient (patient survey) however the average values for those two methods were similar. We did not censor our cohort at the time of renal recovery which was similar to that previously reported in our practice [41]. Since we were not directly estimating survival but rather testing the tools accuracy for predicting death at a certain time point the effect of this should be negligible. Finally our study was limited to a single network and local practice patterns could have introduced some bias. Nonetheless, we closely adhered to the CHARMS recommendations for prognostic validation studies and manual data abstraction from a narrative medical record to supplement electronic pulls of secondary data. Our study adds to the small number of studies assessing prognostic index performance in elderly dialysis patients [18, 19, 42] and also serves as the first external validation of three of the included indices (Wick, Thamer and updated REIN) [19, 24, 29].

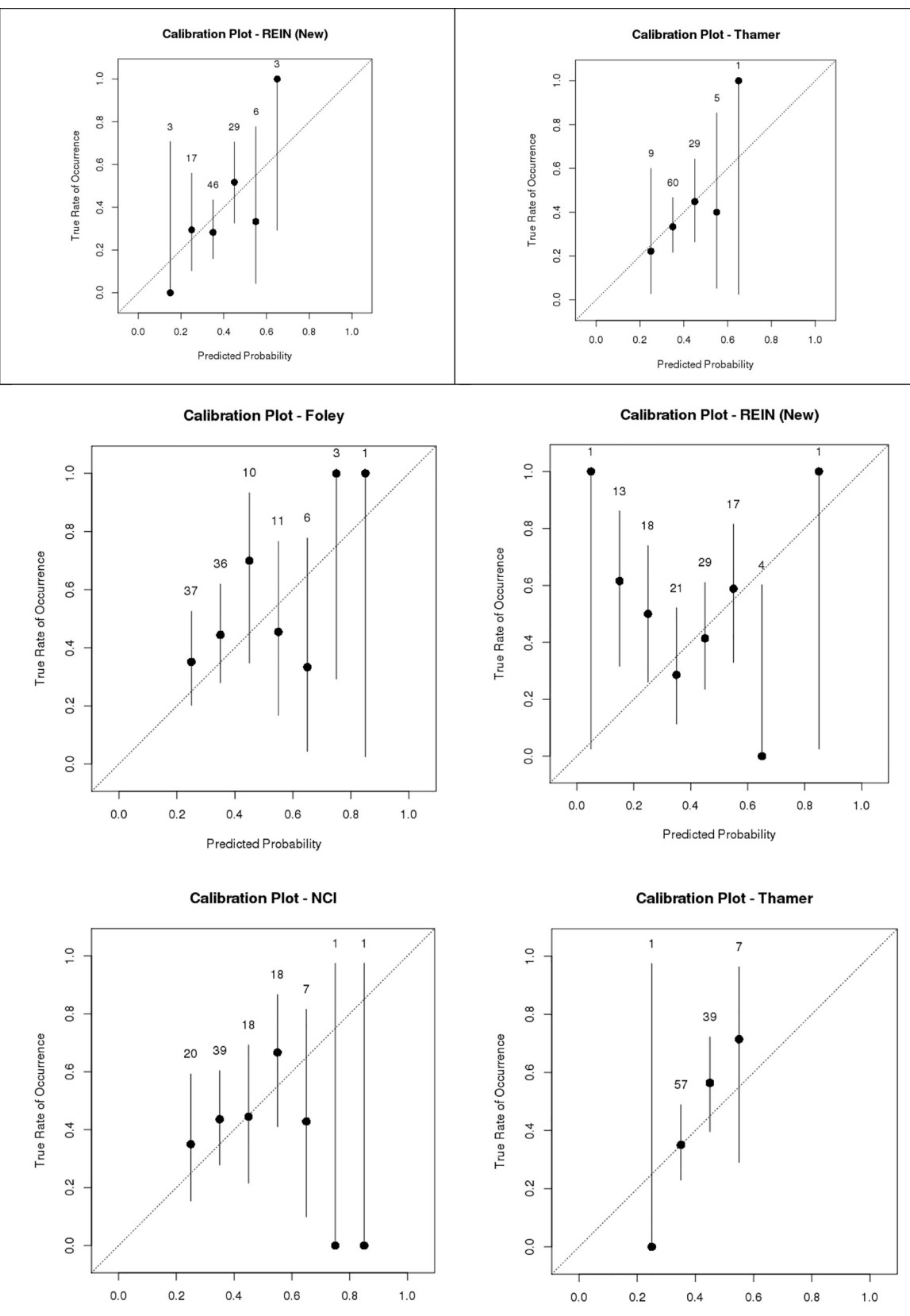

**Fig 4. Calibration plots by index. a.** Mortality at 3 months. **b.** Mortality at 6 months.

Discussion about prognosis and goals of care are especially poignant and relevant for older dialysis patients. The uptake of prognostic indices into clinical practice has been poor with most patients reporting having had no discussions about prognosis at dialysis start [6, 43]. Even for tools that are frequently used in clinical settings (i.e. APACHE III in the ICU) concerns about the ability of prognostic tools to predict accurately for an individual patient lead to a lack of bedside discussions. In qualitative studies, clinicians have expressed skepticism regarding the reliability and accuracy of available tools [9]. Our study confirms that they are justified in their concern. Discrimination between 0.70–0.73 is not sufficient to support high stakes decisions advising on whether to initiate or defer life-sustaining dialysis treatment. Another concern is the wide variation in the gradient of risk (i.e. the percent expected mortality deemed to be "high" by each model), which hinders their interpretability and clinical utility to patients, caregivers, and clinicians. The low positive predictive value for death noted for all the indices was particularly concerning. In fact many of the indices paradoxically had a higher negative than positive predictive value for the highest risk category, a function of the fact that the mortality risk in the highest risk groups in the development cohorts was lower than 50% for many of the indices. The utility of such predictions at the bedside to aid treatment choice thus is questionable, especially when coupled with the absence of being able to predict patient important outcomes for the alternative of no intervention.

Understanding if certain risk thresholds are more or less meaningful to patients and clinicians and how they influence treatment has not been well studied. Furthermore the importance of precision to clinicians and patients in this context also remains unclear.

Even if prognostic indices may not perform well enough on an individual level they may still be acceptable for use on a population level in shaping policy. In particular Medicare coverage in the U.S. limits patients to coverage of either dialysis or hospice, not both as dialysis is considered a life extending treatment. Patients are eligible for the Medicare Hospice benefit if they are deemed more likely than not to die in the next 6 months. The Thamer, Wick and Foley predict more than 50% risk of death within the next 6 months for patients in their highest risk categories. This can support arguments for dual coverage of hospice and dialysis in this high risk group, which in turn could help high risk dialysis patients avoid aggressive and costly treatments that they typically are subject to at the end of life [12, 13].

Developing a more accurate, reliable and generalizable mortality prediction model for older adults facing the decision of whether or not to initiate dialysis may require larger multi-center studies and consideration of a wider array of risk factors including cognitive and functional status, frailty, and social determinants of health [35, 36, 44–47]. Additionally, advanced analytic methods such as machine learning and artificial intelligence, may help identify highest risk patients and facilitate generalizable self-learning models that adapt to each population and setting [48].

## Conclusion

None of the indices performed well in predicting early mortality for the highest risk group in our cohort of elderly incident dialysis patients. The Wick index performed best in terms of discrimination with two other indices, Thamer and Foley having acceptable performance. The future will tell if big data and artificial intelligence can develop more accurate prediction tools but more importantly, better understanding of the role of prognosis at the bedside is needed to promote shared decision making.

## Supporting information

**S1 Checklist. TRIPOD statement.**
(DOCX)

**S1 Table. Variables used to construct Barthel score.**
(DOCX)

**S2 Table. Variables used in current study to construct risk scores.**
(DOCX)

## Author Contributions

**Conceptualization:** Bjorg Thorsteinsdottir, LaTonya J. Hickson, Navdeep Tangri.

**Data curation:** Bjorg Thorsteinsdottir, Rachel Giblon, Atieh Pajouhi, Natalie Connell, Amrit K. Vasdev.

**Formal analysis:** Bjorg Thorsteinsdottir, Rachel Giblon, Megan Branda.

**Funding acquisition:** Bjorg Thorsteinsdottir.

**Investigation:** Bjorg Thorsteinsdottir.

**Methodology:** Bjorg Thorsteinsdottir, Rachel Giblon, Rozalina G. McCoy.

**Supervision:** Megan Branda, Navdeep Tangri, Nilay D. Shah.

**Writing – original draft:** Bjorg Thorsteinsdottir.

**Writing – review & editing:** Bjorg Thorsteinsdottir, LaTonya J. Hickson, Rachel Giblon, Atieh Pajouhi, Natalie Connell, Megan Branda, Amrit K. Vasdev, Rozalina G. McCoy, Ladan Zand, Navdeep Tangri, Nilay D. Shah.

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
