## [Decision Letter · Decision Letter 0]

4 May 2020

PONE-D-20-03399

Validation of prognostic indices for short term mortality in an incident dialysis population of older adults >75

PLOS ONE

Dear Dr. Thorsteinsdottir,

Thank you for submitting your manuscript to PLOS ONE. After careful consideration, we feel that it has merit but does not fully meet PLOS ONE’s publication criteria as it currently stands. Therefore, we invite you to submit a revised version of the manuscript that addresses the points raised during the review process.

We would appreciate receiving your revised manuscript by Jun 18 2020 11:59PM. To enhance the reproducibility of your results, we recommend that if applicable you deposit your laboratory protocols in protocols.io, where a protocol can be assigned its own identifier (DOI) such that it can be cited independently in the future. For instructions see: http://journals.plos.org/plosone/s/submission-guidelines#loc-laboratory-protocols

We look forward to receiving your revised manuscript.

Kind regards,

Davide Bolignano, MD, PhD

Academic Editor

PLOS ONE

"Dr.Tangri reportsgrants andpersonal fees fromAstraZeneca Inc., personal fees from Otsuka Inc., personal fees from Janssen, personal fees from Boehringer Ingelheim and Eli Lilly, grants, and personal fees, and other from Tricida Inc., outside the submitted work"

Please move this information to the Competing Interests section.

Reviewers' comments:

Reviewer's Responses to Questions

**Comments to the Author**

1. Is the manuscript technically sound, and do the data support the conclusions?

Reviewer #1: Partly

Reviewer #2: No

2. Has the statistical analysis been performed appropriately and rigorously? 

Reviewer #1: Yes

Reviewer #2: N/A

3. Have the authors made all data underlying the findings in their manuscript fully available?

Reviewer #1: No

Reviewer #2: No

4. Is the manuscript presented in an intelligible fashion and written in standard English?

Reviewer #1: No

Reviewer #2: Yes

5. Review Comments to the Author

Reviewer #1: Overall, the paper is of interest, considering that there is still debate on the best outcome indicators to be used in clinical practice to assist the difficult decision of whether starting dialysis or not in the elderly.

Compared to other studies testing the role of outcome indicators, this cohort is relatively small. First, this limitation should be underlined more clearly in the abstract and in the text.

Second, given the small sample size, the cohort should be described more precisely (the cohort itself and the population from where the cohort is extracted.

- abstract and results: more information should be given on deaths and their cause and the time frame when they occurred. Moreover, it is of interest to know the mortality rate of the same period occurring in the prevalent population and in younger incident patients.

Introduction and methods: please better introduce and describe the existing prognostic indexes with their pro and contra.

Study-population: please better describe the Mayo Clinic Dialysis Service: how many centers? with which policies? with how many prevalent dialysis patients?

-Research authorisation: was this general or study specific?

-patients: please give information on first dialysis modality (haemodialysis vs peritoneal dialysis). Data are needed also on the first vascular access and on other lab variables (see comments to table1)

-Independent variables: first line there is an extra comma

-Independent variables:"living arrangement" please explain.

Comorbidities were extracted manually: by how many persons? Which was their definition?

-Functional status: this is a key point to be addressed and discussed. First, it was not collected homogeneously, but with two different modalities whether the patient is hospitalised or not. This is important, since by definition hospitalised patients may have a lower functional status. Second, no information is given on how many patients were hospitalised at dialysis start and during follow-up. Finally, the datum is missing nearly in 20% of the cohort. All these considerations need to be addressed considering that, as stated in the discussion, functional status is among the indexes that performed best in this cohort.

-GFR: this is another crucial point. The authors decided to use the Cockroft and Gault formula. The reason of that is to be given, since the most used GFR estimation is nowadays EPI-MDRD formula. Moreover, looking at table 1, half of the cohort had GFR higher or equal to 15 ml/min (i.e. it looks as they started dialysis at stage 4 CKD).

-Baseline data: when were they collected? at the first day of dialysis?

-BMI was missing in 38% of the cases. Please add in the analysis and table the patient body weight (which was used also for GFR estimation, anyway).

- Table 1: i suggest adding the following variables (if some are not available, please add it in the discussion section): Country, GFR estimated also with EPI MDRD, serum creatinine, body weight, vascular access, haemoglobin, CRP, serum phosphate, hospitalisation data, RRT modality. The cohort-specific variables can be added in separate table of "baseline data"

-Is there any information on patients stopping dialysis during the follow up? How were they considered in the analysis?

- A flow diagram describing the cohort should be added.

Reviewer #2: The authors are externally validating 6 new mortality indices (3 of them for the first time) in order to give tools to the clinicians and patients for decision making whether to start dialysis in over 75 year-old patients.

The topic is very interesting, however the main drawback of the study is that the authors did not report which are the characteristics of the indices used in the study and their algorithm (neither in the introduction or methods).

6. PLOS authors have the option to publish the peer review history of their article (what does this mean?). If published, this will include your full peer review and any attached files.

Reviewer #1: Yes: Lucia Del Vecchio

Reviewer #2: No

---

## [Author Response · Author response to Decision Letter 0]

30 Oct 2020

Re: Resubmission of 

PONE-D-20-03399 

Validation of prognostic indices for short term mortality in an incident dialysis population of older adults >75

PLOS ONE

Dear Dr. Davide Bolignano, MD, PhD

Academic Editor

PLOS ONE

We appreciate the review and opportunity to make our paper better for possible publication in 

PLOS ONE. We have carefully addressed all of the reviewers comments as outlined below and evident in our marked up manuscript.

In addition we have formatted our paper to fit the PLOS ONE requirements and uploaded a dataset to Harvard Dataverse.

We do not believe that it is applicable to deposit our laboratory protocols as our research does not involve bench science and the limited standard laboratory results we report come from Mayo Clinic laboratories that are state of the art facilities. Please let us know if you feel differently.

If eligible I would like to opt in for making the peer review history publicly available if eligible. 

1. Please ensure that your manuscript meets PLOS ONE's style requirements, including those for file naming. This has been done, although we were unable to find instructions about how to name the manuscript file. We have named picture and supplemental files as instructed. 

2. Thank you for stating the following in the Financial Disclosure section: "Dr.Tangri reportsgrants andpersonal fees fromAstraZeneca Inc., personal fees from Otsuka Inc., personal fees from Janssen, personal fees from Boehringer Ingelheim and Eli Lilly, grants, and personal fees, and other from Tricida Inc., outside the submitted work"

Please move this information to the Competing Interests section. This has been moved and authors asked to review the PLOS One competing interests and decide if there are other things they want to disclose.

Please confirm that this does not alter your adherence to all PLOS ONE policies on sharing data and materials, This has been done

Please include your updated Competing Interests statement in your cover letter; we will change the online submission form on your behalf. This has been added to the Cover letter and removed from the manuscript draft. 

This has been done

3. Have the authors made all data underlying the findings in their manuscript fully available?

We have prepared a de-identified dataset for sharing

Reviewer #1: No

Reviewer #2: No

Reviewer #1: Overall, the paper is of interest, considering that there is still debate on the best outcome indicators to be used in clinical practice to assist the difficult decision of whether starting dialysis or not in the elderly.

Compared to other studies testing the role of outcome indicators, this cohort is relatively small. First, this limitation should be underlined more clearly in the abstract and in the text.

This has been added to the limitations section in both the abstract and discussion

Second, given the small sample size, the cohort should be described more precisely (the cohort itself and the population from where the cohort is extracted. 

We have described the cohort and created a flow chart for the cohort definition. We also cite studies reporting on a broader cohort from our practice. 

- abstract and results: more information should be given on deaths and their cause and the time frame when they occurred. 

We have reported our 3 and 6 month mortality both in the text and table 1, we do not have detailed cause of death for all of the cohort. 

Moreover, it is of interest to know the mortality rate of the same period occurring in the prevalent population and in younger incident patients.

We have cited studies that report the mortality rate in our younger incident patients, although as reported in methods and discussion our study differs in taking into account also patients who die early from AKI.

Introduction and methods: please better introduce and describe the existing prognostic indexes with their pro and contra.

A description has been added to the methods section however it is too early to make judgement calls on the pro and contra of the different indices as this is part of the motivation for the study.

Study-population: please better describe the Mayo Clinic Dialysis Service: how many centers? with which policies? with how many prevalent dialysis patients?

This information has been added to the methods

-Research authorisation: was this general or study specific?

This has been clarified, the research authorization is general and signed by patients to authorize the use of their records for medical research as per MN law.

-patients: please give information on first dialysis modality (haemodialysis vs peritoneal dialysis). Data are needed also on the first vascular access and on other lab variables (see comments to table1)

These variables have been added where possible and in table 1

-Independent variables: first line there is an extra comma

This has been removed

-Independent variables:"living arrangement" please explain.

This has been added

Comorbidities were extracted manually: by how many persons? Which was their definition? This has been clarified

-Functional status: this is a key point to be addressed and discussed. First, it was not collected homogeneously, but with two different modalities whether the patient is hospitalised or not. This is important, since by definition hospitalised patients may have a lower functional status. Second, no information is given on how many patients were hospitalised at dialysis start and during follow-up. This has been added to table 1 and we did look at functional status for both groups and it was almost identical. In fact the hospital starters were younger and more likely to have AKI than the outpatient starter.s

Finally, the datum is missing nearly in 20% of the cohort. It is unclear as to what you are referring to here? Only 3% were missing functional status. Other variables with missing data needing imputation is outlined in the methods. Please clarify

All these considerations need to be addressed considering that, as stated in the discussion, functional status is among the indexes that performed best in this cohort. 

-GFR: this is another crucial point. The authors decided to use the Cockroft and Gault formula. The reason of that is to be given, since the most used GFR estimation is nowadays EPI-MDRD formula. Moreover, looking at table 1, half of the cohort had GFR higher or equal to 15 ml/min (i.e. it looks as they started dialysis at stage 4 CKD).

I did ask the statistician to go back and look at the formula used to calculate the GFR and it was actually CKD-EPI and not the Cockcroft-Gault. This has now been corrected in the manuscript. A significant number of the patients were acute or acute on chronic kidney failure explaining the high baseline eGFR.

-Baseline data: when were they collected? at the first day of dialysis? This has been clarified

-BMI was missing in 38% of the cases. Please add in the analysis and table the patient body weight (which was used also for GFR estimation, anyway).

- Table 1: i suggest adding the following variables (if some are not available, please add it in the discussion section): Country, GFR estimated also with EPI MDRD, serum creatinine, body weight, vascular access, haemoglobin, CRP, serum phosphate, hospitalisation data, RRT modality. These have been added to table 1 except for CRP as this was available for so few people and not reported in other studies and RRT as this was not reported for any of the other studies, almost all our patients were HD and we did not record whether patients were initiated on IHD or CRRT.

The cohort-specific variables can be added in separate table of "baseline data"

-Is there any information on patients stopping dialysis during the follow up? How were they considered in the analysis? 

We did not censor patients at the time of renal recovery or dialysis cessation. We added to the paper that 60 (17%) of patients stopped dialysis because of renal recovery – this is similar to the rates in our younger incident population as previously published. As mortality 6 months after dialysis initiation was our prognostic endpoint, patients were not censored at the time of dialysis discontinuation and this has been added. This is a valid point brought up by the reviewer, however in our view negligible as we are testing the tools accuracy not directly estimating survival. Censoring those patients isn’t accurate either since it’s a new state (post dialysis remission) and would need to be fully addressed by a multi-state model, which is beyond the scope of this paper.

- A flow diagram describing the cohort should be added. 

This has been added as figure 1

Reviewer #2: The authors are externally validating 6 new mortality indices (3 of them for the first time) in order to give tools to the clinicians and patients for decision making whether to start dialysis in over 75 year-old patients.

The topic is very interesting, however the main drawback of the study is that the authors did not report which are the characteristics of the indices used in the study and their algorithm (neither in the introduction or methods).

We have included more information on the indices included and a reference to our systematic review that underlies our choice of indices.

We hope that you now consider our paper acceptable for publication in your respected journal and appreciate this opportunity to strengthen our paper.

Sincerely

Bjorg Thorsteinsdottir MD

---

## [Editor Report · Decision Letter 1]

3 Dec 2020

Validation of prognostic indices for short term mortality in an incident dialysis population of older adults >75

PONE-D-20-03399R1

Dear Dr. Thorsteinsdottir,

We’re pleased to inform you that your manuscript has been judged scientifically suitable for publication and will be formally accepted for publication once it meets all outstanding technical requirements.

Kind regards,

Davide Bolignano, MD, PhD

Academic Editor

PLOS ONE
---

## [Editor Report · Acceptance letter]

23 Dec 2020

PONE-D-20-03399R1 

Validation of prognostic indices for short term mortality in an incident dialysis population of older adults >75 

Dear Dr. Thorsteinsdottir:

I'm pleased to inform you that your manuscript has been deemed suitable for publication in PLOS ONE. Congratulations! Your manuscript is now with our production department. 

Kind regards, 

on behalf of

Dr. Davide Bolignano 

Academic Editor

PLOS ONE